# From Distribution to Geometry: Stable Graph Generalization via Invariant Barycenters

**Hangyuan Du** [1]  **Rong Wang** [1]  **Weihong Zhang** [1]  **Lu Bai** [2]  **Yu Xie** [1]  **Liang Bai** [3]  **Wenjian Wang** [3,4]

## Abstract

Graph neural networks (GNNs) excel in graph analyzing tasks but often suffer from poor generalization under Out-of-Distribution (OOD) scenarios. Although this problem has attracted increasing attention, most solutions primarily rely on empirical designs, lacking effective mechanisms to characterize and quantify invariance for graph representation learning. To address these limitations, we propose DIGL, a novel graph learning method that improves the OOD generalization of GNNs. Our work makes an initial attempt to geometrize invariance for graphs by introducing computational optimal transport (OT) theory to characterize invariance principle. Specifically, we formulate the underlying invariant prototype shared by graphs across different environments as a distribution barycenter, and consider graph representations in each specific environment as distortions of the prototype. Building on this idea, we establish an invariant learning framework to promote the model to learn purely invariant graph representations for downstream tasks. Moreover, we derive a unified optimization objective for model implementation and provide theoretical analysis to justify our method. Extensive experiments on a broad range of benchmark datasets demonstrate the superior generalization ability of our method compared with baseline methods under various OOD settings.

---

[1]School of Computer and Information Technology, Shanxi University, Taiyuan, China [2]School of Artificial Intelligence, Beijing Normal University, Beijing, China [3]Key Laboratory of Computational Intelligence and Chinese Information Processing (Shanxi University), Ministry of Education, Taiyuan, China [4]Department of Cybersecurity and Protection, Shanxi Police College, Taiyuan, China. Correspondence to: Lu Bai <bailu@bnu.edu.cn>.

*Proceedings of the 43rd International Conference on Machine Learning*, Seoul, South Korea. PMLR 306, 2026. Copyright 2026 by the author(s).

## 1. Introduction

GNNs have emerged as powerful tools for learning from graph data, achieving remarkable success in diverse domains, e.g., scientific discovery (Jumper et al., 2021; Zhong et al., 2025; Shen et al., 2024; Buterez et al., 2024; Liu et al., 2024), computer vision (Liu et al., 2024; Huang et al., 2024; Ma et al., 2024), natural language processing (Fatemi et al., 2023; Wang et al., 2025c) and recommendation systems (He et al., 2020; Chen et al., 2025a; Zhang et al., 2025a). However, their impressive performances heavily rely on the I.I.D. assumption, i.e., the testing and training graph data are independently drawn from an identical distribution (Li et al., 2022). In real-world applications, distribution shifts are usually inevitable and widespread due to the limitation of data collection and uncontrollable data generation (Bengio et al., 2020). Most GNN models fail to generalize to OOD scenarios and often suffer severe performance deterioration, where test distributions differ from training distributions. This vulnerability to distribution shifts stems from a fundamental limitation: GNNs tend to exploit environment-specific spurious correlations present in training data rather than learning truly determinative relationships that generalize across different environments (Wu et al., 2022a; Ju et al., 2026; Li et al., 2025). For instance, in molecular property prediction, a GNN might associate certain functional groups with toxicity since they frequently co-occur in the training dataset, failing to capture the underlying biochemical mechanisms that actually determine molecular property.

Solving the OOD generalization problem is crucial for advancing GNNs toward more real-world applications. Numerous recent studies strive to cope with the challenges through multiple paradigms (Yang et al., 2022). Inspired by Invariant Risk Minimization (IRM), graph invariant learning methods (Li et al., 2022; Wu et al., 2022a; Mo et al., 2024; Wu et al., 2022b; Jia et al., 2024; Sui et al., 2025; Chen et al., 2022; Mao et al., 2026) guide GNNs to identify invariant relationships across different environments by imposing consistency constraints on learned representations. Causal learning approaches (Mo et al., 2024; Gao et al., 2024; Du et al., 2025; Yuan et al., 2025; Fan et al., 2024) seek to uncover the underlying features that causally

determine the graph property. Information bottleneck (IB) based methods (Mao et al., 2026; Yuan et al., 2025; Wu et al., 2020; Sun et al., 2024; Yu et al., 2024; Yang et al., 2024; Di et al., 2025) achieve generalization by preserving sufficient predictive information in the learned representations while compressing task-irrelevant redundant information as much as possible. Graph augmentation methods (Zhu et al., 2024; Sui et al., 2023) enhance distribution diversity to improve model robustness by generating new training instances using perturbation (Wu et al., 2022a) or mixup (Jia et al., 2024) strategies. While these methods have shown promising results in specific applications, they generally promote the model to extract invariant features or structures based on empirical or heuristic designs, leaving a critical unaddressed gap. The absence of appropriate mechanism to characterize and leverage invariance principle in the space of graph distributions significantly limits their effectiveness and reliability. Specifically, existing methods struggle to learn robust graph representations for OOD generalization, facing three fundamental challenges:

*Challenge 1*: Quantification of distribution discrepancy. Distribution shifts in graph data are more intricate than those in Euclidean domains. How to depict and quantify distribution discrepancies across different graph environments remains an open problem.

*Challenge 2*: Characterization of invariance. Although the invariance principle and causal assumption provide conceptual guidance, current methods are often limited to an intuitive level, lacking computationally tractable and optimizable evaluation for invariance of graph data.

*Challenge 3*: Information disentanglement. Due to the scarcity of prior knowledge, it is difficult to disentangle invariant or causal components from environment-specific spurious correlations.

To tackle these challenges, in this work, we propose Distribution Invariant Graph Learning (DIGL), a principled framework that enhances OOD generalization for graph learning model. Different from existing methods, our work explores a novel perspective of characterizing and leveraging invariance mechanism for graph via computational OT theory (Villani, 2008; Titouan et al., 2019). Our core concept is that the graph representations under different environments can be viewed as distributional distortions of a shared, underlying invariant prototype. And the OT Barycenter can be used to estimate this prototype, while the transport process from representations to the barycenter implies the environment-specific information.

Specifically, we leverage the Wasserstein distance to delineate the distributional discrepancy between graphs in multiple environments, providing a rigorous and optimizable metric foundation for learning invariant graph representa-

tions. Then we design a disentanglement strategy to promote the model to separate invariant component of the learned representations from spurious factors, where the former ones are aligned across environments via OT constraints while the latter ones capture environment-specific information. On this basis, we formulate a unified optimization objective for model implementation. Furthermore, we provide theoretical analysis to show the validity of our method. Finally, we conduct a comprehensive set of experiments to verify the effectiveness of our proposed method. The results demonstrate that our DIGL outperforms SOTA baselines under various types of distribution shifts. Our contributions can be summarized as follows.

- We reformulate the invariant mechanism for graph learning by characterizing the geometric structure of graph distributions across multiple environments via OT theory. To the best of our knowledge, this is the first work to perform graph invariant learning from the geometric perspective.

- We propose a novel OOD generalized graph learning method, which learns invariant representations by formulating a unified optimization objective incorporating supervised loss, cross-environment alignment loss, and disentanglement loss. We also provide theoretical analysis to guarantee the validity of the proposed method.

- We test the performance of the proposed method compared with SOTA baseline on diverse graph datasets. The results verify the superiority of our method in improving the OOD generalization.

## 2. Preliminaries

### 2.1. Notations

Let $\mathcal{G}$ and $\mathcal{Y}$ denote sets of input graph and label respectively, $G = (\mathbf{X}, \mathbf{A}) \in \mathcal{G}$ and $Y \in \mathcal{Y}$ represent graph instance and its category label, where $\mathbf{X} \in \mathbb{R}^{N \times D}$ is the attribute matrix and $\mathbf{A} \in \mathbb{R}^{N \times N}$ is the adjacent matrix. The graph predictor $p_\phi(\cdot) = f_\omega(\cdot) \circ g_\theta(\cdot)$ is composed by a GNN encoder $g_\theta(\cdot) : \mathcal{G} \to \mathbb{R}^D$ and a classifier $f_\omega(\cdot) : \mathbb{R}^D \to \mathcal{Y}$. Given training set $\mathcal{G}^{\text{train}} = \{(G_i, Y_i)\}_{i=1}^M$, we evaluate the discrepancy between the prediction and the ground-truth label for graph classification task via a loss function $\ell$ (e.g., cross-entropy loss).

**Problem definition.** OOD generalization refers to the ability to keep high prediction accuracy on unseen test distributions. Given a training set $\mathcal{G}^{\text{train}}$ with the distribution $P^{\text{train}}(G, Y)$, the OOD generalization on graphs aims to learn an optimal graph predictor $p^*$ that can achieve the best prediction performance on testing set $\mathcal{G}^{\text{test}}$, which can

be formulated as:

$$p_\phi^* = \arg\min_{p_\phi} \mathbb{E}_{(G,Y)\in\mathcal{G}^{\text{test}}} \left[\ell\left(p_\phi(G), Y\right)\right], \quad (1)$$

where testing instances are drawn from $P^{\text{test}}(G, Y)$, and $P^{\text{train}}(G, Y) \neq P^{\text{test}}(G, Y)$ indicates the distribution shift between training and testing sets.

## 2.2. Wasserstein Distance

Wasserstein distance is a similarity measure between two probability distributions, which quantifies the minimum total cost required to transform one distribution into another. Given two probability measures $P$ and $Q$ on a compact metric space $(\mathcal{X}, d_X)$, where $\mathcal{X}$ and $d_X$ represent the space and its metric, the 2-Wasserstein distance between $P$ and $Q$ is defined as follows:

$$d_W(P, Q) = \inf_{\pi\in\Pi(P,Q)} \sqrt{\int_{\mathcal{X}\times\mathcal{X}} \|x - y\|_2^2 d\pi(x, y)}, \quad (2)$$

where the coupling $\Pi(P, Q)$ denotes the set of all joint probability measures on $\mathcal{X} \times \mathcal{X}$ with $P$ and $Q$ as margins, i.e., $\int_{\mathcal{X}} \pi(x, y)dx = Q(y)$ and $\int_{\mathcal{X}} \pi(x, y)dy = P(x)$. The Wasserstein distance pursuits the optimal probability measure $\pi^*$, which works as the optimal transport plan between $P$ and $Q$. When $P$ and $Q$ are Gaussian distributions, i.e., $P = \mathcal{N}(\boldsymbol{\mu}_P, \boldsymbol{\Sigma}_P)$ and $Q = \mathcal{N}(\boldsymbol{\mu}_Q, \boldsymbol{\Sigma}_Q)$, we can avoid solving *Eq.* 2 and derive the Wasserstein distance in a closed form:

$$d_W(P, Q) = \Bigg( \left\|\boldsymbol{\mu}_P - \boldsymbol{\mu}_Q\right\|_2^2$$
$$+ \text{tr}\Big(\boldsymbol{\Sigma}_P + \boldsymbol{\Sigma}_Q - 2\big(\boldsymbol{\Sigma}_P^{\frac{1}{2}}\boldsymbol{\Sigma}_Q\boldsymbol{\Sigma}_P^{\frac{1}{2}}\big)^{\frac{1}{2}}\Big) \Bigg)^{\frac{1}{2}}. \quad (3)$$

In the space of probability distribution endowed with the Wasserstein geometry, the Wasserstein barycenter aims to find the geometric mean of a set of probability distributions. Given $m$ probability measures $P_1, \cdots, P_m \in \mathcal{X}$, each of which associated with a weight, denoted as $\lambda_1, \cdots, \lambda_m \in (0, 1)$ with $\sum_{i=1}^m \lambda_i = 1$, the Wasserstein barycenter is the solution to the following minimization problem:

$$B\left(\{P_i\}, \{\lambda_i\}; d_W\right) := \arg\min_B \sum_i \lambda_i d_W^2\left(B, P_i\right). \quad (4)$$

## 3. Method

Inspired by OT theory, we propose a novel invariant learning framework DIGL, whose pipeline is shown in Figure 1, including environment augmentation, subgraph encoding, cross-environment alignment, and representation disentanglement.

### 3.1. Environment Augmentation

Since the testing set is always unseen in the training phase, the optimization of *Eq.* 1 is intractable to implement. In this paper, we follow EERM (Wu et al., 2022a) to mimic the underlying data generation process, using a set of auxiliary environment generators $\mathcal{T} = \{\tau_k\}_{k=1}^T$ to produce environment augmentations $\mathcal{K} = \{\mathcal{G}^k\}_{k=1}^T$ based on $\mathcal{G}^{\text{train}}$. For the specification of $\tau_k$, attribute perturbation (Kong et al., 2022) and structure editing (Wu et al., 2022a) are commonly employed on graphs. In this way, we turn to approximate the following objective via adversarial training:

$$\min_\phi \mathbb{E}_{\mathcal{G}^k\sim\mathcal{K}} \left[\max_{\mathcal{G}^k} \left[\ell_{\text{pre}}\left(p_\phi(G), Y\right) \mid \mathcal{G}^k\right]\right], \quad (5)$$

where the inner loop maximizes the loss to approximate the worst case of $T$ augmented environments.

### 3.2. Subgraph Encoding

In each augmented environment, we separate each graph into invariant subgraph and variant subgraph, and then encode them into latent representations. Since the invariant patterns mainly lies in the local structures of the graph rather than individual nodes, in this paper, we utilize a learnable edge mask (Wu et al., 2022b; Yuan et al., 2025) to extract invariant subgraphs. Given graph $G = (\mathbf{X}, \mathbf{A})$, we employ an initial encoder $g_0$ and a MLP $\varphi$ to generate the mask matrix $\mathbf{M} \in \mathbb{R}^{N\times N}$:

$$\mathbf{h} = g_0(G), \quad \mathbf{W} = \varphi(\mathbf{h}), \quad \mathbf{M}_{u,v} = \mathbf{W}_u \odot \mathbf{W}_v, \quad (6)$$

where $\mathbf{h} \in \mathbb{R}^{N\times d}$ and $\mathbf{W} \in \mathbb{R}^N$ are the learned $d$-dimensional node representations and the corresponding node attentions, respectively, $\mathbf{M}_{u,v}$ denotes the probability that nodes $u$ and $v$ are linked by an edge in the invariant subgraph $\overline{G}$. Then we retain the edges with *Top-$\eta$* probabilities to generate the edge sets of invariant subgraph $\overline{G}$ and variant subgraph $\widetilde{G}$ as follow:

$$\overline{\mathcal{E}} = \text{Top}_\eta(\mathbf{M} \odot \mathbf{A}), \quad \widetilde{\mathcal{E}} = \text{Top}_{1-\eta}((1 - \mathbf{M}) \odot \mathbf{A}), \quad (7)$$

where $\eta$ is a ratio to control the scale of invariant subgraph. With the extracted subgraphs $\overline{G}$ and $\widetilde{G}$, we use two GNN to encode them into node representations $\overline{\mathbf{h}} \in \mathbb{R}^{N\times d}$ and $\widetilde{\mathbf{h}} \in \mathbb{R}^{N\times d}$, and employ a readout function to obtain the subgraph representations $\overline{\mathbf{z}} \in \mathbb{R}^d$ and $\widetilde{\mathbf{z}} \in \mathbb{R}^d$. The process can be formulated as follows:

$$\overline{\mathbf{h}} = g_1(\overline{G}), \quad \widetilde{\mathbf{h}} = g_2(\widetilde{G}),$$
$$\overline{\mathbf{z}} = \text{Readout}(\overline{\mathbf{h}}), \quad \widetilde{\mathbf{z}} = \text{Readout}(\widetilde{\mathbf{h}}). \quad (8)$$

Then, we use a shared classifier $f_\omega$ to produce predictions for subgraph representations $\overline{\mathbf{z}}$ and $\widetilde{\mathbf{z}}$:

$$\overline{Y} = f_\omega(\overline{\mathbf{z}}), \quad \widetilde{Y} = f_\omega(\widetilde{\mathbf{z}}), \quad (9)$$

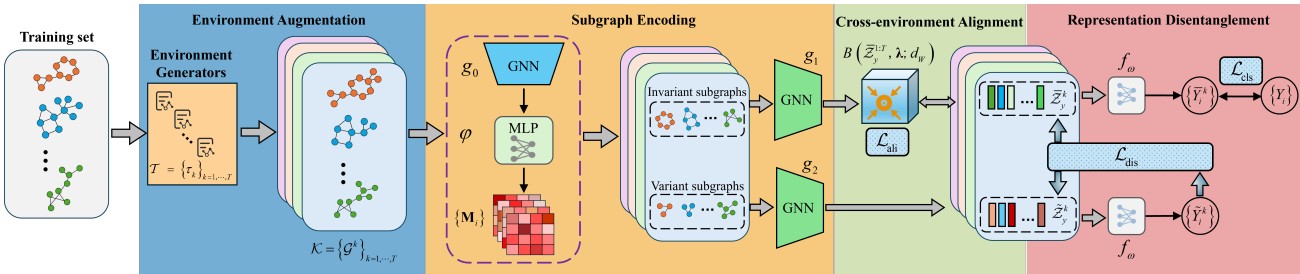

*Figure 1.* The overall framework of DIGL. We employ environment augmentation module to mimic the graph generation processing. In subgraph encoding module, invariant and variant subgraphs are identified and encoded. On this basis, DIGL characterizes the graph invariant prototype across different environments as OT barycenter, and promotes the model to align invariant representations to the prototype while disentangle them from environment-specific information.

where the two predictions will be applied to graph classification and environment adversarial training, respectively. Then we construct the following supervision loss to measure the discrepancy between the predictions and the ground-truth labels across the augmented environments:

$$\mathcal{L}_{\text{cls}} = \sum_{\mathcal{G}^k \sim \mathcal{K}} \sum_{i=1}^{M} \ell\left(\overline{Y}_i^k, Y_i\right), \quad (10)$$

### 3.3. Cross-environment Alignment

To help the encoder generate invariant representations faithfully, we introduce the Wasserstein geometry to characterize the invariant patterns that keep stably under distribution shift. From the geometric view, distributions of graph representations in multiple environments which fall into a same category can be regarded as geometric distortions of one shared underlying category prototype. This prototype carries essential characteristics of the category that are stable to distribution shifts across different environments. As the geometric center of distribution structure defined in metric space, the Wasserstein barycenter provides a powerful tool to estimate the prototype. On this basis, the environment-specific components of each graph can be uncovered in the transport from its representation to the prototype. Figure 2 illustrates our concept.

For the augmented environments in $\mathcal{K}$, we denote the set consisting of graphs in category $y$ as $\left\{\mathcal{G}_y^k\right\}_{k=1}^T$, where $\mathcal{G}_y^k = \left\{G_i^k \mid Y_i = y\right\}_{i=1}^{M_y}$ represents the category of graphs in the $k$-th environment, and $M_y$ is the number of graphs in $\mathcal{G}_y^k$. With the help of the subgraph extraction and encoding module, we encode $\mathcal{G}_y^k$ into two sets of subgraph representations $\overline{\mathcal{Z}}_y^k = \left\{\overline{\mathbf{z}}_i^k \mid Y_i = y\right\}_{i=1}^{M_y}$ and $\widetilde{\mathcal{Z}}_y^k = \left\{\widetilde{\mathbf{z}}_i^k \mid Y_i = y\right\}_{i=1}^{M_y}$, and treat them as samples of a latent distribution defined in the $d$-dimensional space. To simplify the computation, we assume the invariant representations obey a multivariate Gaussian distribution, i.e., $\overline{\mathcal{Z}}_y^k \sim \mathcal{N}\left(\overline{\boldsymbol{\mu}}_y^k, \overline{\boldsymbol{\Sigma}}_y^k\right)$, which can be parameterized by the unbiased

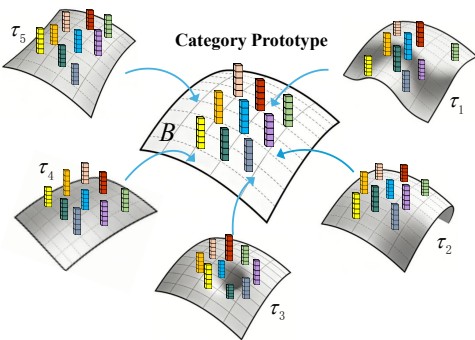

*Figure 2.* An illustration of invariant prototype for a category of graphs. In multiple environment-specific distribution spaces (surrounding surfaces), the graph representations (vectors) in the category have different distribution structures. The central surface indicates the distribution space of the category prototype. And, the blue arrows show the distribution transport from graph representations to their invariant prototype.

estimations of mean and covariance as follows:

$$\overline{\boldsymbol{\mu}}_y^k = \frac{1}{M_y} \sum_{i=1}^{M_y} \overline{\mathbf{z}}_i^k \in \mathbb{R}^d,$$

$$\overline{\boldsymbol{\Sigma}}_y^k = \frac{1}{M_y - 1} \sum_{i=1}^{M_y} \left(\overline{\mathbf{z}}_i^k - \overline{\boldsymbol{\mu}}_y^k\right)\left(\overline{\mathbf{z}}_i^k - \overline{\boldsymbol{\mu}}_y^k\right)^{\mathrm{T}} \in \mathbb{R}^{d \times d}. \quad (11)$$

Then, we estimate the prototype for category $y$ by calculating the Wasserstein barycenter of the sets of invariant representations across different environments:

$$B\left(\overline{\mathcal{Z}}_y^{1:T}, \boldsymbol{\lambda}; d_W\right) := \arg\min_B \sum_k \lambda_k d_W^2\left(B, \overline{\mathcal{Z}}_y^k\right), \quad (12)$$

where the Wasserstein distance is calculated by *Eq. 3*, and the environment weight $\boldsymbol{\lambda} = [\lambda_k] \in \Delta^{K-1}$ is defaulted as uniform. Since invariant representations are expected to be robust against distribution shift across different environments, their distribution structures should be quite similar to the corresponding category prototype. This objective can be achieved by aligning invariant representations in each environment to the prototype, i.e., minimizing the following loss:

$$\mathcal{L}_{\text{ali}} = \sum_{y \in \mathcal{Y}} \sum_{\mathcal{G} \in \mathcal{K}} \lambda_k d_W\left(B\left(\overline{\mathcal{Z}}_y^{1:T}, \boldsymbol{\lambda}; d_W\right), \overline{\mathcal{Z}}_y^k\right). \quad (13)$$

### 3.4. Representation Disentanglement

Imposing only the prototype alignment constraint on $\bar{z}$ is insufficient to ensure the effective disentanglement between invariant factors and environmental information, i.e., the model may still choose to encode environmental information into both $\bar{z}$ and $\widetilde{z}$, leading to degraded solutions. To address the issue, we further adopt adversarial training to explicitly encourage $\widetilde{z}$ to be rich in environmental information, while making $\bar{z}$ as independent of the environment as possible. Specifically, we first introduce the following objective to approximate *Eq.* 5, promoting $\widetilde{z}$ to encode environment information as much as possible:

$$\min - \sum_{\mathcal{G}^k, \mathcal{G}^{k'} \sim \mathcal{K}} \sum_{i=1}^{M} \left\| \widetilde{Y}_i^k - \widetilde{Y}_i^{k'} \right\|_2^2. \tag{14}$$

On this basis, we design the following disentangling regularization to enhance the disentanglement between invariant and variant representations, which encourages $\bar{z}$ and $\widetilde{z}$ to carry complementary information.

$$\min \sum_{\mathcal{G}^k \sim \mathcal{K}} \sum_{i=1}^{M} I\left(\bar{z}_i^k, \tilde{z}_i^k\right), \tag{15}$$

where the upper bound of the mutual information can be estimated by Hilbert-Schmidt Independence Criterion (HSIC) (Song et al., 2007; Gretton et al., 2005) or contrastive loss (e.g. InfoNCE) (Poole et al., 2019; Liu et al., 2023). Then we can derive the following disentanglement objective:

$$L_{\text{dis}} = \sum_{\mathcal{G}^k, \mathcal{G}^{k'} \sim \mathcal{K}} \sum_{i=1}^{M} \left[ I\left(\bar{z}_i^k, \widetilde{z}_i^k\right) - \left\| \widetilde{Y}_i^k - \widetilde{Y}_i^{k'} \right\|_2^2 \right]. \tag{16}$$

This objective can significantly inhibit the "leakage" of environmental information into the invariant representations.

### 3.5. Optimization Process

Taking the above three objectives into account, the overall loss function of DIGL can be formulated as:

$$\mathcal{L}_{\text{DIGL}} = \mathcal{L}_{\text{cls}} + \alpha \mathcal{L}_{\text{ali}} + \beta \mathcal{L}_{\text{dis}}. \tag{17}$$

Update of the category prototype is very challenging: the computations of the Wasserstein distance and the Wasserstein barycenter correspond to two intractable optimization problems, respectively, thus the alignment in *Eq.* 13 is a complicated composition of multiple optimization tasks with different variables. To address such problem, we iteratively update the barycenters and the invariant representations within a batch by alternating optimization. As demonstrated in *Fig.* 3a, for the representation set $\overline{\mathcal{Z}}_y^k$, the feedforward computation of the optimization objective in *Eq.* 13

corresponds to two parts: 1). estimating a category prototype via iteratively solving $T$ optimal transport problems; 2). calculating the alignment error between the prototype and $\overline{\mathcal{Z}}_y^k$. The loopy estimations of the category prototype can be implemented approximately as the recursive update of Wasserstein barycenter. In *Fig.* 3b, we further illustrate the one step calculating in the category prototype estimation, which contains loopy computations of $T$ Wasserstein distance modules.

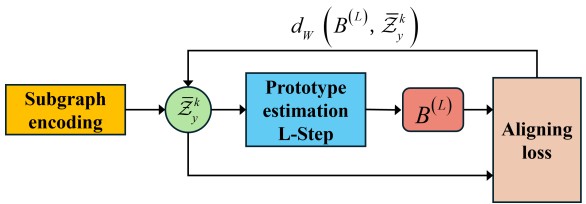

*(a)* The feedforward computation of cross-environment alignment

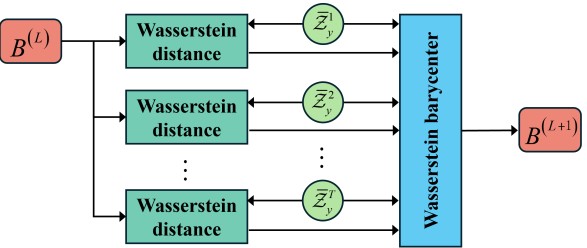

*(b)* One step of the prototype update

*Figure 3.* Alternating optimization of category prototype and invariant representations.

### 3.6. Theoretical Analysis

In this section, we establish the theoretical foundations for our proposed method. We will show that the objective in *Eq.* 17 can provide a valid solution for OOD problem with a generalization risk bound. Given the representation space $\mathcal{Z}$, label space $\mathcal{Y}$, let $f : \mathcal{Z} \to \mathcal{Y}$ be a hypothesis (classifier) from a hypothesis space $\mathcal{H}$, and $\mu$ be a distribution on $\mathcal{Z}$, the expected risk of $f$ is defined as $\mathcal{R}_\mu(f) = \mathbb{E}_{z \sim \mu}[\ell(f(z), y)]$. We assume the classifier $f$ is $K_f$-Lipschitz continuous and the loss function $\ell(\cdot, y)$ is $K_\ell$-Lipschitz continuous with respect to its first argument. Consequently, the composite function $\ell \circ f$ is $K$-Lipschitz continuous with $K = K_f \cdot K_\ell$, i.e., for any $z, z' \in \mathcal{Z}$, there exists $|\ell(f(z), y) - \ell(f(z'), y)| \leq K \|z - z'\|$. According to the Kantorovich-Rubinstein Duality (Villani, 2008), we derive the following proposition bounding the risk difference.

**Proposition 1.** *(Risk Difference Bound). For a $K$-Lipschitz continuous loss-hypothesis composition, the difference in expected risk between any two distributions $\mu$*

and $\mu'$ is bounded by:

$$|\mathcal{R}_\mu(f) - \mathcal{R}_{\mu'}(f)| \leq K \cdot d_W(\mu, \mu').  \quad (18)$$

Building upon this proposition, we demonstrate how DIGL bridges the training and test environments via the estimated prototype.

**Theorem 1.** *(Generalization Risk Bound via Wasserstein Barycenter): For a set of augmented environments $\mathcal{K} = \left\{\mathcal{G}^k\right\}_{k=1}^T$, let $\mu_k$ denote the distribution of graph representations under environment $\mathcal{G}^k$, $B$ denote the Wasserstein barycenter of $\{\mu_k\}_{k=1}^T$, and $\mu_{\text{Test}}$ denote the unseen test distribution, the expected risk $\mathcal{R}_{\text{Test}}(f)$ for any test distribution $\mu_{\text{Test}}$ has the following upper bound:*

$$\mathcal{R}_{\text{Test}}(f) \leq \frac{1}{T}\sum_{k=1}^T \mathcal{R}_{\mu_k}(f) + K\left(\frac{1}{T}\sum_{k=1}^T d_W(\mu_k, B^T)\right) + K \cdot d_W(B, \mu_{\text{Test}}). \quad (19)$$

Detailed proofs of Proposition 1 and Theorem 1 are presented in **Appendix C**. Here, we analyze the connection between the generalization risk upper bound and our learning objective. 1). In *Eq.* 19, the first term denotes the training risk, which is depressed by minimizing the classification loss $\mathcal{L}_{\text{cls}}$ in our objective. 2). The second term means barycenter alignment, which corresponds to our cross-environment alignment loss $\mathcal{L}_{\text{ali}}$. We explicitly minimize the average Wasserstein distance between all the augmented environmental distributions and prototype in our objective. 3). The third term describes the distributional distance between the estimated prototype and the unseen test distribution. Although we cannot directly optimize this term, the prototype is the geometric barycenter learned from the "worst-case" with adversarial augmentation. According to the geometric property, the barycenter tends to be closer to the testing distribution than any training distributions. Thus, the term can be implicitly optimized. In conclusion, our method can guarantee a valid solution for OOD problem by tightening the generalization risk bound via its optimization objective.

# 4. Experiments

In this section, we conduct extensive experiments on a wide range of benchmark datasets to answer the following research questions: **Q1**: How efficient does DIGL improve OOD performance in graph classification task? **Q2**: How does each module contribute to DIGL? **Q3**: Can DIGL effectively identify the invariant components while eliminating spurious features? **Q4**: How do the hyperparameters affect the performance of DIGL.

## 4.1. Experimental Settings

To comprehensively verify the effectiveness of our proposed DIGL, two major categories of graph datasets with different scenarios of distribution shift are utilized to evaluate models OOD generalization performance. Specifically, the first category (Fan et al., 2022) converted from images is disturbed by fixed known data bias and the second category (Gui et al., 2022) derived from the GOOD benchmark contains complex unknown distribution shifts. In experiments, we compare our method with standard Empirical Risk Minimization (ERM) (Vapnik, 1991) and three types of graph invariant learning solutions, as well as two OT based graph learning methods: 1). Augmentation method, including EERM (Wu et al., 2022a), IGM (Jia et al., 2024), and MARIO (Zhu et al., 2024); 2). Causal learning methods, including DIR (Wu et al., 2022b), DisC (Fan et al., 2022), StableGNN (Fan et al., 2024), and GCAL (Zuo et al., 2026); 3). IB principle based methods, including DIVE (Sun et al., 2024), IS-GIB (Yang et al., 2024), IBCS (Yuan et al., 2025), InfoIGL (Mao et al., 2026), and OOD-SEIB (Di et al., 2025). 4). OT based methods, including GDL (Vincent-Cuaz et al., 2021) and GWF-GNN (Xu et al., 2023). Details of our experimental settings are presented in **Appendix D**.

## 4.2. Main Task Results

To answer **Q1**, we compare the graph classification performance of DIGL with that of baselines on the two categories of datasets, respectively. For image-to-graph datasets, we select methods that excel in subgraph extraction and bias mitigation as baselines to handle the spurious substructures, while for GOOD datasets, we adopt more general methods for comparison to evaluate model performance in complex scenarios. The test results are reported in Table 1 and 2, from which we have the following observations.

As shown in Table 1, our DIGL overall outperforms baselines under different bias degrees. As the bias degree increases, all the methods suffer from performance deterioration, however, our DIGL exhibits more prominent superiority. Specifically, under the bias of 0.8, DIGL achieves average improvements of 26.37% compared to the ERM and outperforms the suboptimal model by 3.84% and 7.42% on CMINIST and CKuzushiji, respectively. Under biases of 0.9 and 0.95, DIGL averagely surpasses the ERM and the suboptimal model by 30.95% and 7.09%, respectively. Particularly, although DIGL does not perform best on CFashion under 0.8 degree of bias, it still demonstrates considerable OOD generalization ability compared to the optimal method.

From the results in Table 2, it can be observed that DIGL consistently shows its excellence in most cases. On the contrary, the SOTA methods, e.g., GCAL, IBCS, and In-

*Table 1.* Performances on image-to-graph datasets with different bias degrees.

| Method | CMINIST | | | CFashion | | | CKuzushiji | | |
|---|---|---|---|---|---|---|---|---|---|
| | 0.8 | 0.9 | 0.95 | 0.8 | 0.9 | 0.95 | 0.8 | 0.9 | 0.95 |
| ERM | 61.07±1.58 | 56.32±1.83 | 53.05±2.14 | 59.26±1.39 | 54.77±1.48 | 51.29±1.73 | 48.09±1.21 | 42.59±1.06 | 35.86±1.52 |
| IGM | 67.17±1.13 | 63.42±1.52 | 60.28±1.29 | 61.14±1.42 | 53.28±1.65 | 51.02±1.61 | 50.89±0.92 | 44.36±1.27 | 36.18±1.43 |
| DIR | 23.72±0.55 | 21.27±0.73 | 21.81±0.77 | 32.51±1.87 | 29.72±1.90 | 25.08±1.73 | 28.26±0.35 | 26.46±0.31 | 26.24±0.52 |
| DisC | 80.25 ±3.67 | 74.36±2.03 | 59.76±3.98 | 67.87±1.09 | 59.83±1.28 | 56.09±0.66 | 52.86±1.16 | 43.14±1.31 | 36.25±0.94 |
| GCAL | 78.51 ±3.25 | 77.73±3.08 | 72.90±5.34 | **69.27±0.87** | 61.93±1.24 | 58.64±1.03 | 53.17±1.15 | 49.68±1.49 | 38.13±1.52 |
| DIVE | 75.37±1.84 | 63.29±1.78 | 56.73±1.42 | 67.53±0.59 | 63.66±0.48 | 58.54±0.61 | 50.72±0.89 | 45.38±0.66 | 37.25±1.14 |
| IBCS | 80.54±1.55 | 75.39±1.82 | 64.58±2.16 | 68.27±1.34 | 64.90±1.31 | 57.80±1.76 | 56.18±1.17 | 46.82±1.81 | 36.95±1.67 |
| OOD-SEIB | 79.66±0.89 | 77.26±0.65 | 74.52±1.28 | 68.33±0.79 | 65.29±1.33 | 59.07±1.05 | 54.16±0.93 | 48.81±1.14 | 37.30±1.36 |
| GDL | 72.03±1.36 | 65.24±1.86 | 55.17±1.58 | 57.44±1.22 | 53.42±1.47 | 49.73±1.28 | 48.33±1.06 | 43.57±1.16 | 36.93±1.51 |
| GWF-GNN | 62.76±1.28 | 59.42±1.45 | 51.52±2.07 | 60.46±0.93 | 55.25±1.32 | 51.77±1.56 | 45.53±0.97 | 42.71±1.23 | 38.49±1.48 |
| Ours | **83.63±1.22** | **79.42±1.41** | **78.19±1.27** | 69.14±1.05 | **66.77±0.82** | **64.19±0.95** | **60.35±0.86** | **55.47±1.26** | **43.03±1.18** |

*Table 2.* Performances on GOOD benchmark datasets with covariate and concept shifts.

| Method | Covariate shift | | | | Concept shift | | | |
|---|---|---|---|---|---|---|---|---|
| | HIV Scaffold | SST2 Length | CMNIST Color | Motif Base | HIV Scaffold | SST2 Length | CMNIST Color | Motif Base |
| ERM | 62.05±3.22 | 79.62±1.53 | 28.15±3.36 | 63.57±5.09 | 70.51±3.21 | 72.43±1.88 | 41.95±0.91 | 80.47±0.73 |
| EERM | 69.32±2.72 | 80.36±2.53 | 37.27±3.25 | 77.82±3.27 | 72.18±2.94 | 76.74±1.23 | 43.52±0.78 | 86.19±1.45 |
| MARIO | 68.75±2.04 | 81.74±2.26 | 41.52±2.87 | 75.39±1.95 | 70.85±2.55 | 74.71±0.97 | 44.18±1.13 | 89.39±1.32 |
| DIR | 67.36±1.51 | 77.08±1.72 | 26.37±4.62 | 61.67±5.10 | 68.28±3.77 | 68.76±3.06 | 27.69±3.29 | 78.14±3.94 |
| DisC | 68.31±1.17 | 80.52±2.15 | 44.59±6.63 | 77.94±4.06 | 73.26±2.17 | 75.29±2.18 | 52.09±2.13 | 89.84±1.25 |
| StableGNN | 71.03±1.65 | 80.29±1.95 | 43.28±3.32 | 79.30±3.63 | 71.76±1.90 | 74.37±1.44 | 44.37±1.34 | 88.70±0.73 |
| GCAL | 70.68±1.73 | 82.13±2.36 | 48.37±3.24 | 81.32±3.18 | 73.56±2.61 | 80.40±1.36 | 57.28±1.29 | 90.18±1.33 |
| IS-GIB | 71.75±2.44 | 80.36±2.57 | 38.36±4.02 | 78.79±2.81 | 71.89±2.41 | 73.22±0.82 | 43.59±2.09 | 84.21±1.68 |
| IBCS | 72.55±1.73 | 81.52±1.46 | 45.46±2.71 | **83.15±2.36** | 74.06±2.13 | 76.54±1.57 | 53.92±1.66 | 89.73±1.19 |
| InfoIGL | 71.89±2.19 | 81.87±1.98 | 49.23±3.18 | 79.49±2.73 | 72.88±1.86 | 78.29±1.74 | 50.97±2.21 | 90.74±0.78 |
| GDL | 65.42±2.15 | 75.35±2.02 | 40.62±3.77 | 75.09±2.39 | 72.28±2.17 | 75.32±1.86 | 46.36±2.14 | 85.18±1.21 |
| GWF-GNN | 66.73±2.68 | 72.23±1.94 | 42.09±3.26 | 77.42±2.95 | 71.53±2.25 | 77.16±1.79 | 47.49±1.93 | 82.71±1.54 |
| Ours | **73.38±1.32** | **85.43±1.24** | **53.72±2.83** | 82.52±2.24 | **77.24±1.78** | **83.58±1.24** | **61.88±1.36** | **92.36±0.84** |

foIGL perform unstably when encountering complex types of distribution shifts. That is because these baselines mainly attempt to capture invariant relations or remove spurious features according to empirically or heuristically predefined learning paradigms, lacking an effective mechanism that can characterize the invariance faithfully. Besides, the experimental datasets cover various domains and shift types rather than uniform distribution shift. Consequently, environment-specific information may leak into the learned representations, which makes them fail to identify the real invariant patterns on these datasets. Conversely, our method estimates category prototype from the perspective of geometric structure of distributions and achieves outstanding results, which validates our invariance characterization ability to induce robust representations. Additionally, our disentanglement module further encourages the invariant and variant representations preserve task-determinative information and environment-specific information, respectively, hence demonstrating superior performance under various shifts.

Notably, GDL and GWF both underperform our DIGL in the two scenarios by a large margin. Although OT theory is introduced to measure the distance between graph distributions, these two OT based methods are devoid of an effective mechanism for the handling of distribution shift. As a result, their learned representations cannot produce reliable prediction for the classification task in the OOD scenarios.

### 4.3. Ablation Study

To answer **Q2**, we conduct ablation experiments on GOOD datasets for each module of DIGL. As shown in Table 3, compared with the complete DIGL model, the performances of the three ablated models exhibit different degrees of deterioration, verifying that each module of DIGL is necessary for achieving outstanding OOD generalization ability. Notably, performances of the ablated model without alignment module are generally worse than that of the model removing disentanglement module. This result indicates that our cross-environment alignment objective plays a vital role in improving the OOD generalization ability. Additionally, discarding the disentanglement module also leads to a performance deterioration, since the environmental information cannot be completely eliminated from the invariant representations depend only on the disentangle-

*Table 3.* Results of ablation experiments on GOOD benchmark datasets.

| Method | Covariate shift | | | | Concept shift | | | |
|---|---|---|---|---|---|---|---|---|
| | HIV Scaffold | SST2 Length | CMNIST Color | Motif Base | HIV Scaffold | SST2 Length | CMNIST Color | Motif Base |
| w/o Alignment | 71.31±2.63 | 80.27±3.74 | 39.09±3.61 | 76.52±1.94 | 71.21±2.55 | 73.45±1.62 | 44.64±1.63 | 83.83±1.78 |
| w/o Disentanglement | 71.77±1.68 | 81.70±1.89 | 45.62±2.89 | 79.15±3.27 | 74.02±1.94 | 77.37±1.84 | 50.25±1.46 | 89.76±1.07 |
| w/o (Ali + Dis) | 67.48±2.59 | 79.63±3.02 | 37.60±3.48 | 75.59±2.99 | 70.74±2.34 | 71.05±1.42 | 41.93±1.26 | 83.57±1.58 |
| DIGL | **73.38±1.32** | **85.43±1.24** | **53.72±2.83** | **82.52±2.24** | **77.24±1.78** | **83.58±1.24** | **61.88±1.36** | **92.36±0.84** |

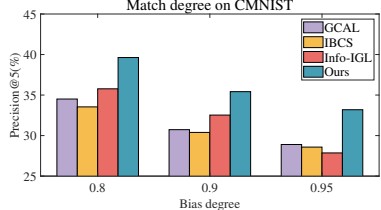 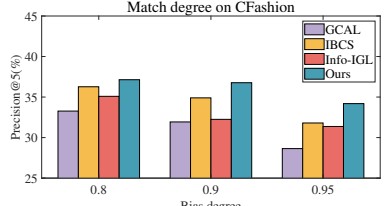 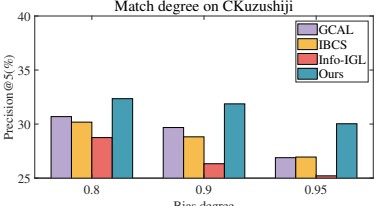

*Figure 4.* Match degree results of different methods on image-to-graph datasets.

ment module until a specialized disentangling mechanism is introduced.

### 4.4. Effectiveness and Interpretability

To answer **Q3**, we leverage precision@5 to measure the match degree between edges of the identified and the ground-truth subgraphs. Compared with SOTA methods, our DIGL consistently achieves higher precision@5 score on image-to-graph datasets with different bias degrees, as illustrated in Figure 4. These results empirically verify that DIGL is capable to identify invariant subgraph more accurately in contrast to baselines, which also reflects our superior interpretable capacity. Additionally, we also visualize the extracted subgraphs of showcases sampled from GOOD-SST2 and GOOD-Motif datasets identified by DIGL and baselines in **Appendix E**.

### 4.5. Sensitivity Analysis

To answer **Q4**, we conduct sensitivity analysis for hyperparameters $\alpha$ and $\beta$ on GOOD-HIV dataset with covariate shift and concept shift, respectively. We adjust $\alpha$ and $\beta$ from 0.1, 0.5, 1, 2, 4 and report AUC results in Figure 5. It can be observed that DIGL remains stable and effective across different hyperparameter settings, validating robustness of our method. Notably, the model reaches the best performance when $\alpha$ gets the large value and $\beta$ is set as a relatively small value. That is because the alignment objective is essential for capturing invariant factors from the input graph, a large $\alpha$ can encourage the learned invariant representations under different environments to be aligned to their category prototypes. As for $\beta$, a comparatively smaller value will further enhance the generalization ability by preventing environmental information leaking into the learned invariant representations.

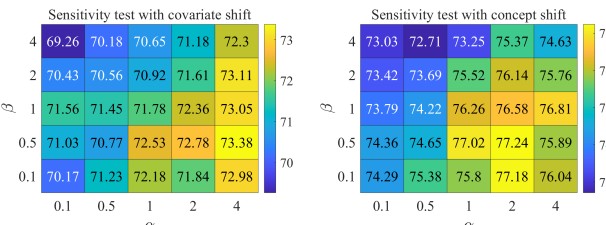

*Figure 5.* Sensitivity test on GOOD-HIV for hyperparameters.

## Conclusion

In this paper, we explore to learn invariant graph representations from a geometric perspective, which can endow the GNN model outstanding OOD generalization ability against distribution shifts. Based on this concept, we consider that the distributions of graph representations in a same category share an invariant geometric prototype and develops a novel DIGL model consisting of four tailored modules. We conduct comprehensive experiments on several graph benchmark datasets to show the superiority of our method over state-of-the-art baselines against diverse distribution shifts. This work contributes to bridging the gap in understanding and characterizing invariant learning principle for graph data with the help of computational optimal transport theory, providing valuable insights for future research in this field.

In future work, we plan to devise a more advanced implementation strategy to further improve the applicability of our DIGL in large-scale scenarios. The potential solution may lie in introducing lightweight GNN architectures for encoder and leveraging approximate estimations for Wasserstein distance. Additionally, we would like to extend our work to node and edge tasks, as well as explainability for GNNs.

## Impact Statement

This paper presents work whose goal is to advance the field of Graph Machine Learning. There are many potential societal consequences of our work, none which we feel must be specifically highlighted here.

## Acknowledgments

This work is supported by the National Natural Science Foundation of China (62576198, 62576371, 62476157, 62472270), the Humanity and Social Science Foundation of Ministry of Education (24YJAZH022), the Key Research and Development Program of Shanxi Province (202302010101007).

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

# Appendix

The content of the appendix is organized as follows:

- Appendix **A** provides detailed related works in complementary to that in Introduction.

- Appendix **B** provides the training algorithm and computational complexity analysis for DIGL.

- Appendix **C** includes the proofs for all the theorems in the main content.

- Appendix **D** details the experimental settings, including dataset description, evaluation protocols, baselines, and reproducibility.

- Appendix **E** demonstrates additional experimental results.

## A. Related Work

### A.1. Graph OOD Generalization

Driven by the growing need for handling distribution shifts widely existed in real-world graph applications, graph OOD generalization has become a particularly critical problem. Several works attempt to address the problem from various perspectives, including data generation (Wu et al., 2022a; Yang et al., 2022; Zhu et al., 2024; Sui et al., 2023), architecture design (Sui et al., 2025; Yuan et al., 2025; Guo et al., 2024), and learning principle (Wu et al., 2022b; Chen et al., 2022; Mao et al., 2026; Fan et al., 2024; Zhang et al., 2025b). Meanwhile, some others strive to improve models generalization for distribution shifts in certain applications, e.g., financial risk modeling (Li et al., 2025), molecular prediction (Yang et al., 2022), healthcare (Huang et al., 2023), and recommendation systems (Zhao et al., 2025). Among these works, graph invariant learning provides a general principle for achieving OOD generalization based on its practical assumption, which guides the model to exploit the invariant relationships between input graphs and labels across distribution shifts, while removing the variant spurious correlations.

Following the invariant learning principle, a surge of recent methods have been emerging for pursuing invariant relationships from input graphs. EERM (Wu et al., 2022a), IGM (Jia et al., 2024), MARIO (Zhu et al., 2024), and GOLD (Wang et al., 2025b) leverage data augmentation strategy to detour the scarcity of environment annotations, enforcing GNNs to identify invariant patterns that are more stably related with the labels across different environments. Backed by causal theory, DIR (Wu et al., 2022b), GCIL (Mo et al., 2024), CIGA (Chen et al., 2022), DisC (Fan et al., 2022), and GCAL (Zuo et al., 2026) propose to learn invariant subgraphs or representations by enhancing the exploitation of causal features that fundamentally determine the labels while suppressing the extraction of non-causal information during the learning process. InfoIGL (Mao et al., 2026), DIVE (Sun et al., 2024), IS-GIB (Yang et al., 2024), and OOD-SEIB (Di et al., 2025) introduce the information bottleneck objective to preserve maximal task-relevant information as invariance for prediction while compressing redundant information to eliminate spurious features and noise. These invariant learning methods strive to enhance OOD generalization by learning certain patterns or relationships that exhibit stability while facing distribution shifts. However, they characterize invariance in an empirical or intuitive way, failing to ensure the model to faithfully exploit invariant relationships for prediction.

### A.2. Optimal Transport on Graphs

As a well-established research field, OT defines a metric between distributions, which provides a principled tool to understand the space of probability distributions. Given the cost that evaluates the distance between two distributions, OT can be interpreted as moving the mass from the source distribution to the target one with minimal expected total cost (Villani, 2008). Over the past years, OT has been employed to solve different problems in machine learning (Fatras et al., 2022; Shafieezadeh-Abadeh et al., 2019; Arjovsky et al., 2017; Tolstikhin et al., 2018; Vincent-Cuaz et al., 2021; Kolouri et al., 2018; Courty et al., 2017; Metelli et al., 2019), because of its capacity to compare and manipulate probability distributions. For graph analysis, Gromov-Wasserstein (GW) distance (Mémoli, 2011) and its enhanced version Fused Gromov-Wasserstein (FGW) distance (Titouan et al., 2019) offer two practical pseudo-metrics for measuring the distance between two graphs. Meanwhile, several algorithms for approximating these OT distances with low complexity, including Sinkhorn scaling (Cuturi, 2013), Bregman ADMM (Wang & Banerjee, 2014), Inexact Proximal Point (Xie et al., 2020), and Conditional Gradient (Titouan et al., 2019), have been proposed in succession.

Recently, a serious of works extensively investigate extensions of OT for graph learning tasks. GOT (Petric Maretic et al., 2019), RESAlign (Chen et al., 2025b) and HLOT (Yan et al., 2025) adopt OT theory to develop alignment frameworks for the challenging problem of graph matching under supervised, semi-supervised, and unsupervised settings, respectively. GDL (Vincent-Cuaz et al., 2021) introduces GW divergence as the data fitting term to encode graphs through their nodes pairwise relations and model them as convex combination of graph atoms. The GWF in (Xu et al., 2023) establishes a flexible and interpretable graph representation learning paradigm, which inherits the interpretability of factorization models by representing each graph via a weighted combination of graph factors under the GW discrepancy. To eliminate the inconsistency in GNNs loss functions introduced by the non-i.i.d problem of node labels, QW loss (Cheng & Xu, 2024) proposes a Quasi-Wasserstein distance between the observed multi-dimensional node labels and their estimations, and converts the label prediction to an optimal label transport problem defined on graph edges. WatE (Cheng et al., 2025) develops a Wasserstein t-distributed stochastic neighbor embedding method, leading to an information-enriched learning framework for graph visualization. To avoid generation of inconsistent sample-label pairs in graph mixup, GEOMIX (Zeng et al., 2024) proposes to employ equivalence-preserving transformations to align graphs in the GW space and interpolates graphs on the GW geodesics, ensuring the consistency between mixup samples and labels.

In this work, we take advantage of OT theory to characterize invariant relationships from a geometric perspective, aiming at tackling the challenges that existing methods have faced. To the best of our knowledge, this is the first attempt to explore such a characterization for addressing graph OOD generalization problem.

## B. Algorithm and Complexity Analysis

The overall training procedure of the proposed DIGL is summarized in Algorithm 1.

The computational cost of DIGL mainly derives from four parts: environment augmentation, subgraph encoding, cross-environment alignment, and representation disentanglement, where the dominant computational load stems from the GNN encoders and the cross-environment alignment. For the graphs with $N$ nodes, the propagation consumption of GNN encoder per iteration is generally $O(Nd^2)$, where $d$ is the representation dimension. As for the alignment model, the cost is mainly from the update of Wasserstein barycenter. The computation of Wasserstein distance between distributions of graph representations in two different environments contributes $O(d^3)$ time complexity. As a result, the update of category prototype across $T$ augmented environments needs $O(Td^3)$ computational cost per iteration. Based on the analysis above, the overall computational cost of DIGL is $O(Nd^2 + Td^3)$, which is mainly determined by the architecture of GNN encoders in large scale graph tasks since $N \gg T \cdot d$. Fortunately, many lightweight GNN models, e.g., LightGCN (He et al., 2020), LightGNN (Chen et al., 2025a), and GSAT(Wang et al., 2025a), can be equipped into DIGL as the encoder to reduce the computational cost. Additionally, some approximate estimation methods for Wasserstein distance, e.g., low-rank covariance approximation and Monte Carlo sampling approximation can further reduce the complexity of our model training, which is left as one of our future work.

---

**Algorithm 1** The training procedure of DIGL

---

**Input** : A set of training graphs $\{G_i, Y_i\}_{i=1}^M$ and all given hyper-parameters;
**Output** : Environment generators, Subgraph extractor, encoders, and classifier;
 1: Parameter initialization;
 2: **while** not convergence **do**
 3:     Generate a set of environment augmentations $\mathcal{K} = \left\{\mathcal{G}^k\right\}_{k=1,\cdots,T}$ based on $\mathcal{G}^{\text{train}}$ ;
 4:     In each augmented environment, learn invariant representation $\bar{\mathbf{z}}_i^k$ and variant representation $\tilde{\mathbf{z}}_i^k$ for each graph using *Eq.* 8;
 5:     Align learned invariant representations to corresponding category prototypes;
 6:     Perform disentanglement for invariant and variant representations;
 7:     Produce prediction using invariant representations;
 8:     Calculate the loss $\mathcal{L}_{\text{DIGL}}$ with *Eq.* 17;
 9:     Update model parameters to minimize the loss.
10: **end while**
11: **return** $result$

---

## C. Proof

**Proposition 1.** *(Risk Difference Bound). For a $K$-Lipschitz continuous loss-hypothesis composition, the difference in expected risk between any two distributions $\mu$ and $\mu'$ is bounded by:*

$$|\mathcal{R}_\mu(f) - \mathcal{R}_{\mu'}(f)| \leq K \cdot d_W(\mu, \mu').$$

*Proof.* Since $\ell(z) = \ell(f(z), y)$ is $K$-Lipschitz continuous, we can rewrite $\ell(z) = K \cdot \hat{\ell}(z)$ where $\hat{\ell}(z)$ is 1-Lipschitz continuous. Then we have:

$$
\begin{aligned}
|\mathcal{R}_\mu(f) - \mathcal{R}_{\mu'}(f)| &= |\mathbb{E}_{z \sim \mu}[\ell(z)] - \mathbb{E}_{z' \sim \mu'}[\ell(z')]| \\
&= K \cdot \left| \mathbb{E}_{z \sim \mu}[\hat{\ell}(z)] - \mathbb{E}_{z' \sim \mu'}[\hat{\ell}(z')] \right| \\
&\leq K \cdot \sup \left( \mathbb{E}_{z \sim \mu}[\hat{\ell}(z)] - \mathbb{E}_{z' \sim \mu'}[\hat{\ell}(z')] \right).
\end{aligned}
\tag{20}
$$

According to the Kantorovich-Rubinstein Duality theorem (Villani, 2008), the supremum of the expectation difference over 1-Lipschitz functions is exactly equal to the following 1-Wasserstein distance:

$$\sup \left( \mathbb{E}_{z \sim \mu}[\hat{\ell}(z)] - \mathbb{E}_{z' \sim \mu'}[\hat{\ell}(z')] \right) = d_{W_1}(\mu, \mu'), \tag{21}$$

where $d_{W_1}(\cdot, \cdot)$ denotes the 1-Wasserstein distance between two distributions. Since the inequality $d_{W_1}(\mu, \mu') \leq d_{W_2}(\mu, \mu')$ holds true due to the monotonicity of $L_p$ norms, we can obtain the final upper bound:

$$|\mathcal{R}_\mu(f) - \mathcal{R}_{\mu'}(f)| \leq K \cdot d_W(\mu, \mu').$$

$\square$

**Theorem 1.** *(Generalization Bound via Wasserstein Barycenter): For a set of augmented environments $\mathcal{K} = \{\mathcal{G}^k\}_{k=1}^T$, let $\mu_k$ denote the distribution of graph representations under environment $\mathcal{G}^k$, $B$ denote the Wasserstein barycenter of $\{\mu_k\}_{k=1}^T$, and $\mu_{\text{Test}}$ denote the unseen test distribution, the expected risk $\mathcal{R}_{\text{Test}}(f)$ for any test distribution $\mu_{\text{Test}}$ has the following upper bound:*

$$\mathcal{R}_{\text{Test}}(f) \leq \frac{1}{T} \sum_{k=1}^T \mathcal{R}_{\mu_k}(f) + K \left( \frac{1}{T} \sum_{k=1}^T d_W(\mu_k, B) \right) + K \cdot d_W(B, \mu_{\text{Test}}).$$

*Proof.* Consider the relationship between the expected risks on test environment and any single augmented environment $\mathcal{G}^k$, the following triangle inequality holds:

$$\mathcal{R}_{\text{Test}}(f) \leq \mathcal{R}_{\mu_k}(f) + |\mathcal{R}_{\text{Test}}(f) - \mathcal{R}_{\mu_k}(f)|, \tag{22}$$

By applying **Proposition** 1, we obtain the upper bound of the risk difference term via the Wasserstein distance:

$$\mathcal{R}_{\text{Test}}(f) \leq \mathcal{R}_{\mu_k}(f) + K \cdot d_W(\mu_k, \mu_{\text{Test}}). \tag{23}$$

This inequality holds for any single augmented environment, thus we can average it over all augmented environments:

$$\mathcal{R}_{\text{Test}}(f) \leq \frac{1}{T} \sum_{k=1}^T \mathcal{R}_{\mu_k}(f) + \frac{K}{T} \sum_{k=1}^T d_W(\mu_k, \mu_{\text{Test}}). \tag{24}$$

By substituting the triangle inequality of the Wasserstein metric $d_W(\mu_k, \mu_{\text{Test}}) \leq d_W(\mu_k, B) + d_W(B, \mu_{\text{Test}})$ into *Eq.* 24, we have:

$$
\begin{aligned}
\mathcal{R}_{\text{Test}}(f) &\leq \frac{1}{T} \sum_{k=1}^T \mathcal{R}_{\mu_k}(f) + \frac{K}{T} \sum_{k=1}^T [d_W(\mu_k, B) + d_W(B, \mu_{\text{Test}})] \\
&= \frac{1}{T} \sum_{k=1}^T \mathcal{R}_{\mu_k}(f) + K \left( \frac{1}{T} \sum_{k=1}^T d_W(\mu_k, B) \right) + K \cdot d_W(B, \mu_{\text{Test}}).
\end{aligned}
\tag{25}
$$

$\square$

# D. Details of Experimental Settings

## D.1. Datasets and Evaluation Protocols

In this subsection, we introduce the detailed information for experimental datasets and our evaluation protocols.

**Fixed bias degrees on image-based datasets.** The first category of datasets includes three datasets: CMNIST, CFashion, and CKuzushiji (Fan et al., 2022). Superpixel graphs in these datasets are transformed from images via the KNN method with at most 75 nodes each graph using. For training data, samples with three bias degrees (i.e., 0.8, 0.9, and 0.95) are constructed, where each category highly correlates with a pre-defined background color. In testing sets, the unbiased samples are used to evaluate the models OOD generalization performance. Each graph is labeled by its original digit class, so that its digital subgraph is deterministic for label and background subgraph is spuriously correlated with labels but not deterministic. Details of these datasets are presented in Table 4.

*Table 4.* Details for datasets: CMNIST, CFashion, and CKuzushiji.

| Dataset | #Avg. Nodes | #Avg. Edges | #Classes | #Graphs (train/val/test) | Label-relevant subgraph | Bias degree | Difficulty |
|---|---|---|---|---|---|---|---|
| CMNIST | 61.09 | 488.78 | 10 | 10K/5K/10K | Digit | 0.8/0.9/0.95 | Easy |
| CFashion | 61.03 | 488.26 | 10 | 10K/5K/10K | Fashion product | 0.8/0.9/0.95 | Medium |
| CKuzushiji | 52.87 | 423.0 | 10 | 10K/5K/10K | Hiragana | 0.8/0.9/0.95 | Hard |

**Complex distribution shifts on GOOD datasets.** In the second category of datasets, both covariate shift and concept shift are considered (Gui et al., 2022). Four datasets derived from the GOOD benchmark are selected in our experiments, whose details are shown in Table 5.

- GOOD-HIV is a small-scale real-world molecular dataset, where molecular graphs are constructed by treating atoms and chemical bonds as nodes and edges, respectively. The task is to predict whether the molecules have the property of inhibiting HIV virus replication.

- GOOD-SST2 is a real-world natural language sentimental analysis dataset. In this sentiment binary classification dataset, sentences or text sequences are converted to sentiment graphs using BERT and Biaffine parser, where nodes denote words, and edges indicate their relations.

- GOOD-CMNIST is also an image-to-graph dataset similar to CMNIST. The difference is that GOOD-CMNIST has no fixed bias degree and contains complex distribution shift types such as covariate shift and concept shift.

- GOOD-Motif is a synthetic dataset motivated by Spurious-Motif, where each graph in the dataset is generated by connecting a base graph and a motif, and the label is determined by the motif solely. To study covariate and concept shifts separately, the base graph type and the size are selected as domain features.

*Table 5.* Details for GOOD benchmark datasets.

| Dataset | Covariate shift | | | | | Concept shift | | | | |
|---|---|---|---|---|---|---|---|---|---|---|
| | Train | ID val | ID test | OOD val | OOD test | Train | ID val | ID test | OOD val | OOD test |
| GOOD-HIV | 24,682 | 4112 | 4112 | 4113 | 4108 | 15,209 | 3258 | 3258 | 9365 | 10,037 |
| GOOD-SST2 | 24,744 | 5301 | 5301 | 17,206 | 17,490 | 27,270 | 5843 | 5843 | 15,142 | 15,944 |
| GOOD-CMNIST | 42,000 | 7000 | 7000 | 7000 | 7000 | 29,400 | 6300 | 6300 | 14,000 | 14,000 |
| GOOD-Motif | 18,000 | 3000 | 3000 | 3000 | 3000 | 12,600 | 2700 | 2700 | 6000 | 6000 |

**Evaluation protocols.** With the consideration of data characteristics and task objectives, we use AUC metric for the GOOD-HIV dataset while employing classification accuracy (ACC) for other datasets. There is significant label imbalance in the GOOD-HIV dataset, i.e., positive samples only account for a very small proportion of the total molecular graphs. AUC measures models global ability to distinguish between positive and negative samples and is insensitive to the category distribution ratio. In contrast, other datasets have more balanced category distributions, thus ACC can evaluate models predictions on these datasets directly and intuitively.

## D.2. Baselines

In order to evaluate the proposed DIGL, we compare its performance on the graph classification task with that of the following baselines.

- ERM (Vapnik, 1991): An intuitional learning paradigm that trains model by directly minimizing the average loss on the training set.

- EERM (Wu et al., 2022a): An invariant learning framework that introduces multiple adversarially trained context explorers to maximize the variance of risks from multiple virtual environments.

- IGM (Jia et al., 2024): A co-mixup learning method that jointly generates mixed multiple environments and captures invariant patterns from the mixed graph data.

- MARIO (Zhu et al., 2024): An adversarial graph augmentation method that integrates invariant learning and IB principle for improving OOD generalizability of graph contrastive learning models.

- DIR (Wu et al., 2022b): The method identifies the invariant causal reasoning across different distributions by performing intervention on the training distribution, so as to improve the generalizability and intrinsic interpretability of GNNs.

- DisC (Fan et al., 2022): A debiasing framework for GNNs via learning disentangled causal substructure, which explicitly filters edges into causal and bias subgraphs by a parameterized edge mask generator and designs a casual-aware loss and a bias-aware loss to supervise two corresponding GNN modules respectively.

- StableGNN (Fan et al., 2024): A general causal representation learning framework for GNNs, which jointly learns the high-level graph representations with the causal variable distinguisher.

- GCAL (Zuo et al., 2026): The model employs the graph attention mechanism to explicitly divide the input graph into causal and shortcut subgraphs and introduces information theory to learn corresponding disentangled representations.

- InfoIGL (Mao et al., 2026): A multi-level contrastive learning framework based on IB theory, which introduces a redundancy filter to compress environment-relevant factors and enhance models generalization ability to unseen distributions.

- DIVE (Sun et al., 2024): The framework trains a collection of models to focus on all label-predictive subgraphs by encouraging these models to foster divergence on the subgraph mask, circumventing the limitation of solely focusing on the subgraph corresponding to simple structural patterns.

- IS-GIB (Yang et al., 2024): A unified framework to address OOD problem, which proposes individual GIB to remove spurious feature by minimizing the mutual information between the input graph and its embeddings and designs structure GIB to leverage the structural intra- and inter-domain correlations.

- IBCS (Yuan et al., 2025): With the hypothesis that graphs are composed of causal, spurious, and noisy subgraphs, the model separate spurious and noisy information from the causal subgraph with the help of causal subgraph objective and IB principle.

- OOD-SEIB (Di et al., 2025): Aiming at traceably capturing the structural distribution changes between input graph and extracted subgraph, the method employs structural entropy to measure the inherent information changes for better OOD generalization.

- GDL (Vincent-Cuaz et al., 2021): An online graph dictionary learning approach based on GW divergence, which encodes graphs through their nodes pairwise relations and models them as convex combination of dictionary elements.

- GWF-GNN (Xu et al., 2023): The model establishes an explicit factorization mechanism for graphs under the GW discrepancy and represents each graph by the weights of graph factors, where each factor reflects a representative pattern hiding in the data, and each weight denotes the similarity between the graph and a factor.

## D.3. Implementation Details

In our experiments, we adopt GIN (Xu et al., 2019) as the backbone, which has the layer number $L = 4$ and representation dimension $d = 128$. We set the number of augmented environments as $T = 10$, and set the maximum ratio of invariant subgraph as $\eta = 0.3$. For all models in experiments, we train each model with the Adam optimizer, and set the learning rate as 0.01, the batch-size as 128, and the training epoch as 200. The hyperparameters in all baselines are set according to their authors' suggestion. In our method, we employ Mean-Pooling to obtain entire graph representations on each dataset, and leverage 3-layer and 2-layer MLPs with ReLU activation function as the subgraph extractor and the classifier, respectively. Our experiments are performed on NVIDIA A800 80G with the running configurations and packages including Ubuntu 22.04, CUDA 11.8, Python 3.10, and PyTorch 2.1.

## E. Supplementary Experiments

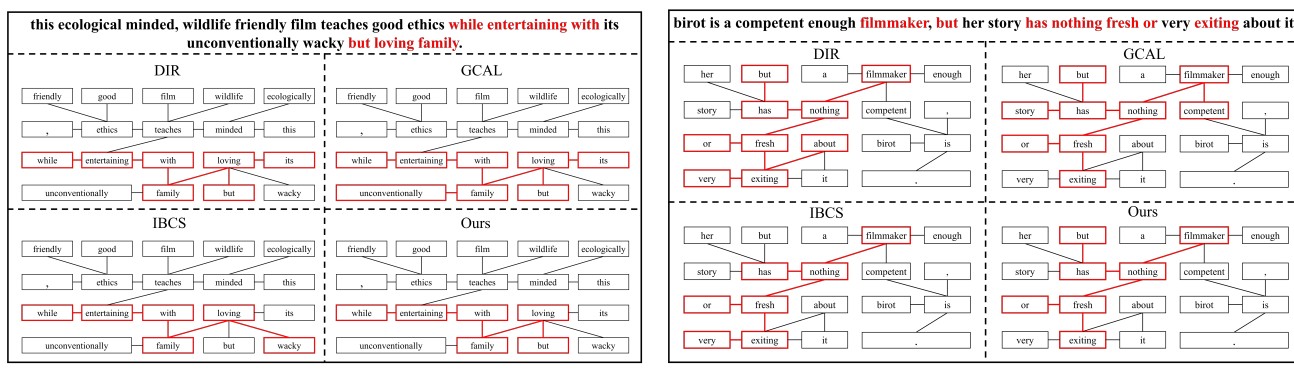

*(a)* Sentence with positive sentiment polarity

*(b)* Sentence with negative sentiment polarity.

*Figure 6.* Visualizing the identified label-relevant subgraphs of showcases from GOOD-SST2.

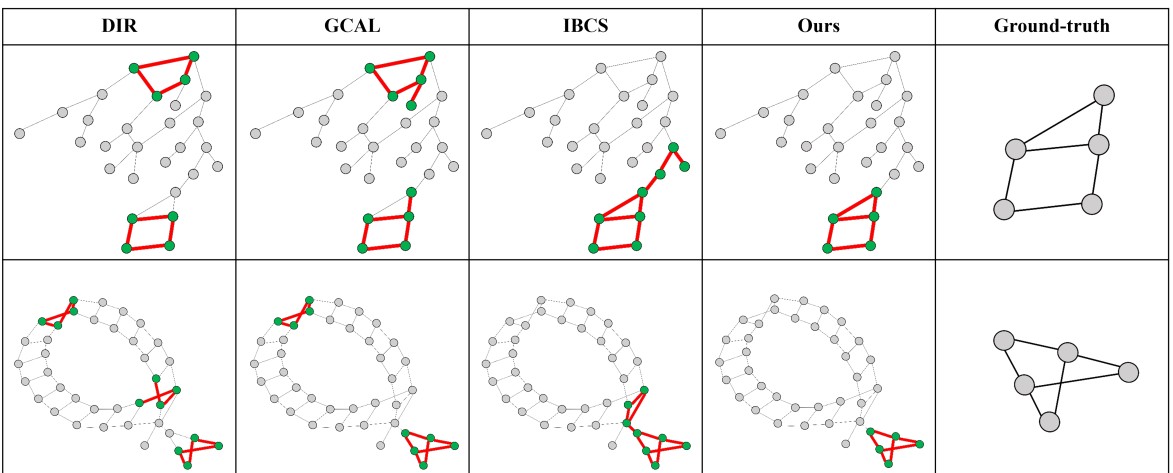

*Figure 7.* Visualizing the identified label-relevant subgraphs of showcases from GOOD-Motif.

In this section, we provide additional experimental results and analysis to supplement the results provided in the main text.

### E.1. Case Visualization

To intuitively present the advantages of our method over baselines in capturing causal information, we visualize the identified subgraph of showcases in GOOD-SST2 and GOOD-Motif by DIGL and baselines. In Figure 6, we select two film reviews from GOOD-SST2, which are correctly classified as positive and negative, respectively. We mark the important words that determine the sentiment polarity of the sentences in red color, and compare the corresponding subgraphs extracted from the sentence graphs by different methods. It can be observed from the figure that our method adeptly identifies

the pivotal words. In contrast, other baselines either fail to completely identify entire determining words or tend to capture redundant words. In Figure 7, two graphs from GOOD-Motif dataset are used for testing, whose categories are determined by the specific motifs. We can obtain the observation that the baselines always extract invariant subgraphs along with some spurious structures, while our DIGL accurately identifies label-relevant subgraphs that match the ground-truth. The results verify the superiority of our method in eliminating spurious correlation, yielding better generalization and interpretability for graph classification.

### E.2. Convergence Analysis

In the implementation of our DIGL, two-fold optimization is adopted in model training, which encourages invariant representations of different environments to align to their category prototypes according to the feedforward computation of the optimization objective, and estimates invariant prototype for each category based on the updated representations. To verify the convergency and stability of our training process, we present the total training loss of each epoch on CMINST dataset with biases of 0.8. 0.9, and 0.95 during the training stage, as illustrated in Fig 8. It can be clearly observed that the training loss of DIGL can reduce to a relatively low level on the CMNIST dataset with different data biases after several epochs. Furthermore, we plot the mean loss after 80 epochs with dashed lines to intuitively demonstrate the convergence. Observations reveal that the training loss converges and hovers within a narrow margin of the mean value after 80 epochs. This evidence supports the viability of our optimization mechanism, which contributes to favorable convergence and stable training.

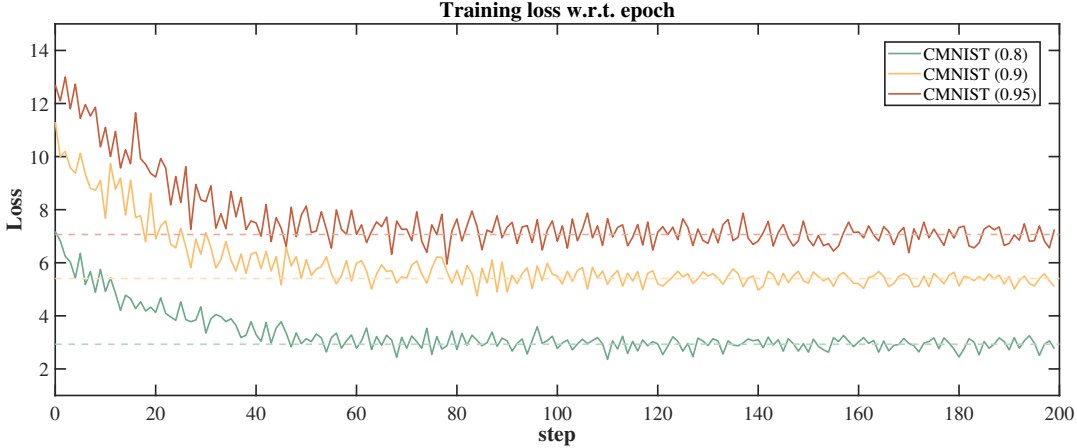

*Figure 8.* The loss of each epoch on CMNIST during the training process.

### E.3. Efficiency Analysis

*Table 6.* Comparison of training time (s).

| Method | Dataset | | | | | | |
|---|---|---|---|---|---|---|---|
| | CMNIST | CFashion | CKuzushiji | GOOD-HIV | GOOD-SST2 | GOOD-CMNIST | GOOD-Motif |
| ERM | 17.64 | 17.13 | 14.72 | 22.59 | 35.16 | 40.04 | 19.35 |
| EERM | 45.06 | 43.28 | 37.48 | 48.36 | 60.21 | 67.57 | 46.91 |
| DIR | 31.76 | 31.14 | 25.43 | 42.75 | 44.39 | 58.12 | 33.87 |
| MARIO | 58.32 | 57.27 | 46.81 | 68.47 | 76.03 | 88.30 | 63.48 |
| IBCS | 35.19 | 32.67 | 28.58 | 51.78 | 58.43 | 65.39 | 48.98 |
| InfoIGL | 33.62 | 31.83 | 28.74 | 52.39 | 57.76 | 68.25 | 47.63 |
| GDL | 53.47 | 52.19 | 48.35 | 66.57 | 78.08 | 85.64 | 59.53 |
| GWF-GNN | 63.05 | 60.74 | 49.26 | 72.19 | 85.24 | 89.66 | 62.37 |
| Ours | 47.32 | 47.06 | 38.53 | 50.24 | 62.33 | 68.09 | 47.91 |

In this subsection, we perform efficiency analysis to test whether our DIGL achieve outstanding results with incurring significant computational costs. Under the uniform setting of training environment and training hyper-parameters, we compare the training time to assess the efficiency of DIGL and baselines. In Table 6, we report the training time of one epoch for each method. The results demonstrate that our DIGL endures the comparable or moderately longer training time compared with related baselines, and can simultaneously induce pronounced performance improvement according to Tables 1 and 2. It is worth noting that our method costs shorter training time in contrast to the two OT based models GDL and GWF-GNN, as well as the augmentation method MARIO. It shows both the excellent efficiency and the effectiveness of our model and further verifies the computational feasibility of DIGL. We can conclude that our DIGL can strike a balance between OOD generalization performance and computational efficiency.

