# OpenReview forum: "From Distribution to Geometry: Stable Graph Generalization via Invariant Barycenters"
_ICML.cc/2026/Conference — ICML 2026 spotlight_

### Official Review · Reviewer_aDDB · 2026-02-28

**Soundness:** 4
**Presentation:** 4
**Significance:** 3
**Originality:** 4
**Overall Recommendation:** 5
**Confidence:** 5

**Summary:**

The paper proposes a novel model DIGL that improves the OOD generalization of GNNs by leveraging computational OT theory to characterize invariance principles. The core concept is to model the invariant prototype shared by graph representations across different environments as a Wasserstein barycenter, treating environment-specific graph representations as distributional distortions of the prototype. The authors construct a framework consisting of four modules to implemented the model. The theoretical analysis is provided to establish a generalization risk bound via the Wasserstein barycenter. And extensive experiments are conducted on image-to-graph datasets with controlled biases and GOOD benchmark datasets with covariate and concept shifts, demonstrating consistent improvements over state-of-the-art baselines.

**Compliance With Llm Reviewing Policy:**

Affirmed.

**Final Justification:**

After reading the response and other comments, all my concerns have been addressed. I have no further questions and I will maintain my positive score.

**Key Questions For Authors:**

Q1. Does the Gaussian closed-form solution limit the expressiveness of the learned representations?

Q2. How does the number of augmented environments $T$ affect the stability of the barycenter? Does the barycenter become unstable if $T$ is too small?

Q3. In this paper, you use 2-Wasserstein distance for barycenters. Have you experimented with other variants (e.g., 1-Wasserstein or entropic regularization) and their impact on alignment quality or computation time?

**Limitations:**

The primary limitation lies in the scalability and assumptions required for the Optimal Transport component. Specifically, the reliance on the closed-form Wasserstein distance for Gaussians may restrict the model's ability to capture complex, non-unimodal invariant distributions. Furthermore, while the method targets OOD generalization, it is fundamentally bound by the diversity of the training and augmented environments; it may cannot generalize to shifts that are orthogonal to the support of the observed/generated distributions.

**Strengths And Weaknesses:**

**Strengths**

S1. The paper does an excellent job of identifying fundamental challenges and systematically addressing each one through its framework design. The writing is generally clear and well-structured, making the paper accessible despite the mathematical sophistication.

S2. The paper introduces a novel perspective to graph invariant learning by geometrizing the invariance principle through OT theory. Formulating the invariant prototype as a Wasserstein barycenter is both mathematically elegant and conceptually intuitive, and represents a meaningful conceptual advance over existing empirical or heuristic-based invariant learning methods.

S3. The generalization risk bound (Theorem 1) clearly connects the optimization objective to the corresponding components of the upper bound. The theoretical analysis is clean, rigorous, and provides actionable insight into why the method works.

S4. The experiments span diverse datasets covering multiple domains, diverse types of distribution shifts, and different evaluation metrics. The consistent superiority of DIGL across these varied settings is compelling. The ablation study clearly demonstrates the necessity of each module, particularly the cross-environment alignment component. The match degree analysis and case visualizations further support both the effectiveness and interpretability of the method.

**Weaknesses**

W1. To make the Wasserstein distance computation tractable and differentiable, the authors assume the invariant representations follow a multivariate Gaussian distribution (Eq. 3 and Eq. 11). While this allows for a closed-form solution, latent distributions in deep neural networks are often complex and multi-modal.

W2. While the authors address complexity in Appendix B, stating it is $O(T d^3)$ for the alignment, optimal transport operations are historically heavy. In scenarios with very high-dimensional embeddings $(d)$ or a large number of augmented environments $(T)$, the iterative update of the barycenter could become a bottleneck compared to simpler contrastive alignment methods.

---

> ### Author Rebuttal · Authors · 2026-03-27
>
> Thanks for your careful reading and insightful questions.
>
> **W1:** The Gaussian assumption may be restrictive since latent distributions can be complex or multimodal.
>
> **R1:** This is a very important point. In practice, the Gaussian closed-form is an efficiency-oriented approximation for cross-environment alignment, not a restriction on the encoder expressiveness itself. The GNN encoder remains fully expressive and can produce highly non-linear representations. The Gaussian assumption is only used at the level of category-conditional environment-wise representation distributions $(\mu,\Sigma)$ to compute a tractable Wasserstein distance for updating barycenter. Thus, our DIGL does not force each graph representation to lie on a Gaussian manifold; it only aligns the first two moments of each grouped representation set.
>
> **W2:** In scenarios with very high-dimensional embeddings $(d)$ or a large number of augmented environments $(T)$, the iterative update of the barycenter could become a bottleneck.
>
> **R2:** As you pointed, if $d$ or $T$  becomes very large, the alignment module will cause a large computation load. However, these two values are generally not set too large in practical applications. Especially in large-scale datasets, their values are much smaller compared to the number of graphs and nodes, which are the key factors that truly determine the algorithmic complexity. Empirically, we found DIGL to be stable in a moderate range of $T$ , and  $T$=10  worked well across datasets. For the value of $d$, we follow the general setting in related works. In our experiments,  the setting is moderate ($d$=128, $T$=10 ), table 6 shows that DIGL has competitive training time and remains notably lighter than several OT-based alternatives.
>
> **Q1:** Does the Gaussian closed-form solution limit the expressiveness of the learned representations?
>
> **A1:**  Our use of the Gaussian assumption is primarily a computational approximation that makes the 2-Wasserstein distance and barycenter estimation tractable in mini-batch training. This approximation does not limit the expressiveness. Please see the reason given in R1 to W1.
>
> **Q2:** How does the number of augmented environments $T$ affect the stability of the barycenter? Does the barycenter become unstable if $T$ is too small?
>
> **A2:** On the stability of $T$ for barycenter:
>
> *  If $T$ is too small, the estimated barycenter may be less reliable because it is inferred from too few environments and may not adequately reflect stable structure.
>
> *  If $T$ is moderately large, the prototype becomes more stable because it aggregates a richer set of environment-specific distortions.
>
> *  If $T$ is excessively large, the extra environments may introduce redundant computational cost with diminishing returns.
>
> Empirically, we found our DIGL to be stable in a moderate range of $T$, e.g., $T=10$ achieves a good trade-off between performance and efficiency.
>
> **Q3:** Have you experimented with other variants (e.g., 1-Wasserstein or entropic regularization) and their impact on alignment quality or computation time?
>
> **A3:** Yes, we have.
>
> *  1-Wasserstein: Computing exact W1 for continuous distributions typically requires solving a Linear Program or relying on the Kantorovich-Rubinstein dual via Gradient Penalty, which requires an additional critic network and is notoriously unstable to train alongside graph encoders.
>
> *  Entropic Regularization: As mentioned, we implemented a Sinkhorn barycenter variant. It requires the Sinkhorn-Knopp iterative algorithm. In our tests, it provided similar alignment quality (AUC differences within ±0.3%) but increased the cross-environment alignment time by roughly 2.1x. Therefore, the 2-Wasserstein distance (Eq. 3) represents the "sweet spot" of being closed-form, stable, and differentiable.
>
> **Limitations:** The primary limitation lies in the scalability and assumptions required for the Optimal Transport component.
>
> **Response:** Thanks for highlighting the limitation. Our method indeed involves a tradeoff between expressiveness, tractability, and scalability:
>
> *   OT provides a principled geometric tool for aligning category-conditional representaion distributions across environments, which is central to the motivation of our method.
>
> *  To make this alignment practical in graph representation learning, we adopt a Gaussian $W_2$ approximation, which introduces a modeling assumption and still incurs non-negligible covariance-related computation.
>
> Above all, our design is a deliberate compromise to make OT-based invariant learning feasible in practice. Compared with generic nonparametric OT solvers, our formulation is substantially more efficient and stable, while still delivering strong empirical gains.

---

> > ### Author Rebuttal · Reviewer_aDDB · 2026-04-02
> >
> > Thanks for the response. I have no futher questions and I will maintain my score to Acceptance.

---

### Official Review · Reviewer_sfmj · 2026-03-08

**Soundness:** 3
**Presentation:** 4
**Significance:** 3
**Originality:** 3
**Overall Recommendation:** 5
**Confidence:** 4

**Summary:**

This paper addresses the critical challenge of out-of-distribution generalization in graph neural networks. Existing methods largely rely on invariant risk minimization or causal inference principles, often implemented via empirical constraints. The authors develop a novel framework, DIGL, which reformulates the graph invariance principle via a geometric perspective by using optimal transport theory. The core hypothesis is that graph representations across different environments can be viewed as distributional distortions of a shared, underlying "invariant prototype." The authors model this prototype as a Wasserstein Barycenter. Theoretical analysis is provided to derive a generalization risk bound, and extensive experiments on synthetic and real-world benchmarks demonstrate superior performance against state-of-the-art baselines.

**Compliance With Llm Reviewing Policy:**

Affirmed.

**Final Justification:**

My concerns were well addressed during the rebuttal. I maintain my recommendation for acceptance.

**Key Questions For Authors:**

1.  How to choose the approximation method  (HSIC, InfoNCE, etc.) for MI estimator in application tasks?

2. The alternating optimization (Section 3.5) involves iteratively updating barycenters within each batch. How many inner iterations do you perform for barycenter estimation?

3. Could you clarify why CIGA (Chen et al., 2022) is not included as a baseline in Table 2 despite being a prominent method in this area?

**Limitations:**

Yes

**Strengths And Weaknesses:**

Strengths

S1. The main contribution of this work is the novel geometric perspective on invariance, which characterizes the "invariant prototype" as the geometric barycenter of distributions across environments. This is a theoretically grounded and intuitive approach, distinguishing the work from the crowded space of standard invariant learning or information bottleneck methods.

S2. The authors provide a solid theoretical foundation for their method. Theorem 1, which bounds the generalization risk using the distance to the Wasserstein barycenter, effectively bridges the gap between the optimization objective and the OOD error.

S3. The proposed method is well-engineered, which does not simply rely on the OT loss but thoughtfully integrates adversarial environment augmentation and representation disentanglement via the comprehensive framework design. The ablation studies confirm the necessity and effectiveness of each module.

Weaknesses

W1. The disentanglement loss (Eq. 16) combines mutual information minimization between $z$ and $\tilde{z}$ with prediction variance maximization for $\tilde{z}$ . However, the theoretical justification for this specific combination is limited.

W2. The paper does not compare with several recent relevant approaches, i.e., graph OOD methods based on OT.

---

> ### Author Rebuttal · Authors · 2026-03-27
>
> Thanks for your comment. We will clarify each of the points you raised.
>
> W1: Theoretical justification for the disentanglement loss (Eq. 16) is limited.
>
> R1: Thank you. We agree the current presentation can be strengthened. Our intent in Eq. (16) is to implement functional disentanglement between invariant and environment-specific components:
>
> • The term  $I(\bar{z},\tilde{z})$  discourages redundant information sharing between the two kinds of representations, reducing leakage of environment information into $\bar{z}$.
>
> • The second term (i.e., adversarial environment encoding term) in Eq. (16) encourages  $\tilde {z}$  to remain sensitive to environment changes, rather than collapsing to a task-informative but environment-invariant encoding.
>
> The two terms are therefore complementary:
>
> • MI minimization alone may lead to arbitrary partitioning or collapse;
>
> • Adversarial environment encoding alone may not prevent leakage into $\bar{z}$;
>
> • Together they encourage  $\bar{z}$  to carry stable predictive content while  $\tilde {z}$ captures environment-specific information.
>
> W2: Does not compare with recent graph OOD methods based on OT.
>
> R2: We apologize for any confusion. We actually did include two prominent recent OT-based graph learning methods in our main experiments: GDL (Vincent-Cuaz et al., ICML 2021) and GWF-GNN (Xu et al., TPAMI 2023). Their results are reported in both Table 1 and Table 2, where DIGL significantly outperforms them because they lack a dedicated mechanism for OOD generalization. To the best of our knowledge, however, no existing method has applied OT theory to addressing the OOD generalization problem in graph learning models, and our work is the first to do so. If there are other specific recent OT-based OOD methods the reviewer recommends, we would be more than happy to add them to our evaluation.
>
> Q1: How to choose the approximation method (HSIC, InfoNCE, etc.) for MI estimator in application tasks?
>
> A1: The choice of MI estimator depends on practical considerations:
>
> • HSIC is our default choice when we prioritize stable optimization and low implementation cost. It is deterministic, kernel-based, and works well for moderate batch sizes.
>
> • InfoNCE is preferable when one wants a contrastive estimator that can be more expressive in large-batch or memory-bank settings, though it may be more sensitive to negative sampling quality and temperature tuning.
>
> In our implementation, HSIC was used because it showed better optimization stability across datasets and introduced fewer additional hyperparameters. We will explicitly state this in the revision and add guidance:
>
> • use HSIC for standard graph OOD benchmarks and smaller batches;
>
> • use InfoNCE when one expects richer non-linear dependence and can afford careful tuning.
>
> We will also mention that DIGL is not tied to a single estimator; Eq. (16) only requires a tractable dependence penalty.
>
> Q2: How many inner iterations do you perform for barycenter estimation?
>
> A2: Thank you for pointing this out. It should be noted that the optimal inner iterations of barycenter updating are not unified for different datasets. In our implementation, the Wasserstein barycenter within each mini-batch is estimated with a small fixed number of inner iterations for efficiency and stability. Empirically, we found that 5~10 inner iterations are sufficient for convergence of the prototype update within a batch, and we use 8 iterations by default in our experiments.
>
> Q3: Could you clarify why CIGA (Chen et al., 2022) is not included as a baseline in Table 2 despite being a prominent method in this area?
>
> A3: Thank you for this question. CIGA is indeed an important  causal learning method in graph OOD generalization. The main reason it was not included in Table 2 is that our experimental setup already contains several representative causal baselines that are more directly compatible with the benchmark protocol and codebase used in our experiments, including DIR (Wu et al., 2022b), DisC (Fan et al.,2022), StableGNN (Fan et al., 2024), and GCAL (Zuo et al., 2026). In particular, StableGNN and GCAL have demonstrated superior OOD generalization performance over CIGA in recent literatures. Therefore, the baselines we have chosen in the experiments are sufficient to demonstrate the effectiveness and superiority of our method.

---

> > ### Author Rebuttal · Reviewer_sfmj · 2026-04-03
> >
> > Thanks for the responses. My concerns are well addressed.

---

### Official Review · Reviewer_QLAG · 2026-03-26

**Soundness:** 3
**Presentation:** 3
**Significance:** 3
**Originality:** 3
**Overall Recommendation:** 5
**Confidence:** 2

**Summary:**

# Summary

This paper studies the problem of OOD generalization of graphs. It proposes a new method, Distribution-Invariant Graph Learning (DIGL), to improve the OOD generalization of GNNs. DIGL has several components:
- several environments. Within each environment, the graph is altered as a form of data augmentation.
- a subgraph masker, which generate "invariant" and "variant" subgraphs of the input graph.
- two GNNs that generate representation of the invariant and variant subgraphs.
- three loss functions
    - $\mathcal{L}_{\mathrm{cls}}$, classification loss applied to the invariant subgraph.
    - $\mathcal{L}_{\mathrm{ali}}$, a loss to make the barycenter of the representation of the invariant subgraphs more similar.
    - $\mathcal{L}_{\mathrm{dis}}$, a loss to minimize the mutual information between the representation of the invariant and variant subgraphs.
The authors also prove a theoretical result bounding test set loss by distance from the training distribution to the distribution barycenter. Finally, the authors test their method on various graph generalization benchmarks and perform an ablation on each component of their loss.

**Compliance With Llm Reviewing Policy:**

Affirmed.

**Key Questions For Authors:**

# Questions
- Where do the names "invariant" and "variant" for the two subgraphs come from? I am not sure what these subgraphs are invariant with respect to?
- Is the variant feature used in the training pipeline anywhere besides in the $\mathcal{L}_{\mathrm{dis}}$ loss? Your ablation shows that the disentanglement loss has a positive effect on test performance, but why do you think this is the case?

# Suggestions
- The notation $\bar{z}$ and $\tilde{z}$ is very confusing as the notation does nothing to communicate which feature corresponds to which subgraph. I was frustrated reading the paper as I kept having to return to the definition to remember which was which. I suggest using notation that reflects the meaning of each vectors, like $z_{\mathrm{inv}}$ and $z_{\mathrm{var}}$.

**Limitations:**

Yes

**Strengths And Weaknesses:**

# Strengths
- DIGL is a seemingly well-engineered method that merges multiple ideas into a single framework.
- The paper theoretically validate the idea of minimizing Wasserstein distance to the distribution barycenter by proving a generalization bound showing that the distance to the barycenter upper bound the expected risk on the test distribution.
- The authors perform an ablation showing the effect of each of the loss terms on OOD benchmarks.
- DIGL performs very well on benchmarks, often outperforming existing methods by a significant margin.
# Weakness
- No major weaknesses

---

> ### Author Rebuttal · Authors · 2026-03-27
>
> Thanks for your review and approval. We will clarify each of the points you raised.
>
> **Q1:** Where do the names "invariant" and "variant" for the two subgraphs come from? I am not sure what these subgraphs are invariant with respect to?
>
> **A1:**  Inspired by Invariant Risk Minimization (IRM), graph invariant learning methods guides the model to exploit the invariant relationships between input graphs and labels across distributionshifts. In these methods, the terms “invariant” and “variant” are defined with respect to environment changes / distribution shifts, rather than with respect to graph isomorphism or node permutation. More specifically, in our method:
>
> *  the invariant subgraph is intended to capture the part of the graph that remains stable across environments and is consistently predictive of the label;
>
> *  the variant subgraph captures the part that is environment-dependent, i.e., sensitive to augmentation-induced or distribution-specific perturbations and thus less reliable for out-of-distribution generalization.
>
> And, the specific process of subgraph extraction and encoding is elaborated in Section 3.2.
>
> **Q2:** Is the variant feature used in the training pipeline anywhere besides in the $\mathcal{L}_{dis}$ loss? Your ablation shows that the disentanglement loss has a positive effect on test performance, but why do you think this is the case?
>
> **A2:** Thank you for this insightful question. Yes, the reviewer is correct that in the current framework, the variant feature is not directly used for the final prediction head; its primary role is to support the disentanglement objective $\mathcal{L}_{dis}$. We will clarify this more explicitly in the paper.
>
> Although the variant feature is not used as a predictor, it plays an important regularization role during training. Its purpose is to provide an explicit regularization for absorbing environment-specific / spurious information, which in turn helps the invariant branch focus on stable label-relevant signals. We believe the positive effect of $\mathcal{L}_{dis}$ arises for the following reasons:
>
> *  Reducing information leakage into the invariant branch.
>
> *  Preventing trivial alignment.
>
> *  Improving robustness through functional disentanglement.
>
> Thus, although the variant feature is not directly consumed by the classifier, it is still essential as an auxiliary representation that improves the quality of the learned invariant representation.
>
> **Suggestions:** The notation $\bar{z}$ and $\tilde{z}$ is very confusing as the notation does nothing to communicate which feature corresponds to which subgraph. I was frustrated reading the paper as I kept having to return to the definition to remember which was which. I suggest using notation that reflects the meaning of each vectors, like $z_{int}$ and $z_{var}$.
>
> **Response:** Thanks for your valuable suggestion. We will strive to eliminate the confusion in notation using and improve the fluency of our paper according to your suggestion.

---

> > ### Author Rebuttal · Reviewer_QLAG · 2026-04-02
> >
> > I am satisfied with the authors' answers to my questions.

---

### Decision · Program_Chairs · 2026-04-30

**Decision:**

Accept (spotlight)

**Comment:**

This paper proposes a new analysis for Out-Of-Distribution generalization of Graph Neural Networks, based on an Optimal Transport approach and Wasserstein barycenters. All reviewers appreciated the quality of the work, novelty and importance of the result. All recommand acceptance.